# Cognitive Function, and Its Relationships with Comorbidities, Physical Activity, and Muscular Strength in Korean Older Adults

**DOI:** 10.3390/bs13030212

**Published:** 2023-03-01

**Authors:** Shinuk Kim

**Affiliations:** Department of Smart Information Communication Engineering, Sangmyung University, Cheonan 31066, Republic of Korea; kshinuk@gmail.com; Tel.: +82-041-550-5452

**Keywords:** chronic conditions, physical activity, muscular strength, cognition, older adults

## Abstract

Background: Little is known regarding how much physical activity (PA) and lower-body muscle strength (LBMS) together can help to reduce the negative effect of comorbidities on cognitive function. This study examined the moderating effects of PA and LBMS in determining the relationship between comorbidities and cognitive function in older Korean adults. Materials and Methods: This is a population-based cross-sectional study. Data for this study were taken from the 2020 Korea Longitudinal Study on Aging (KLoSA) in South Korea using a computer-assisted personal interview. The 2020 KLoSA survey included a total of 10,097 older individuals aged 65 and older (6062 females and 4035 men). Comorbidities were determined based on physician-diagnosed chronic conditions. PA and LBMS were measured with a self-reported questionnaire and a sit-to-stand test, respectively. Cognitive function was assessed using the Korean version of the Mini-Mental Status Examination for dementia screening. Results: Multimorbidity was correlated with an increased risk (odds ratio, OR = 1.415, *p* < 0.001) of cognitive impairment. Insufficient PA and weak LBMS were correlated with an increased risk of cognitive impairment (OR = 1.340, *p* < 0.001; OR = 1.719, *p* < 0.001, respectively). Particularly, PA modulates the negative impact of comorbidities on cognitive function (β = −0.3833; 95% CI = −0.4743 to −0.2023) independent of all measured covariates. Weak LBMS was found to be an independent predictor of cognitive function (β = −2.5078, *p* < 0.001) regardless of comorbidities. Conclusions: The study findings suggest that a lifestyle intervention targeting regular PA and muscular fitness should be a therapeutic means against cognitive decline associated with normal aging and/or chronic diseases.

## 1. Introduction

Population aging is a worldwide phenomenon that reflects a result of both falling birth rates and increasing life expectancy [1]. South Korea has seen the fastest increase in the number of elderly persons among all nations, from 7.2% of the population older than 65 years in 2000 to 15.5% of the population older than 65 years in 2017 [2]. A sudden rise in the elderly population raises the possibility of a range of medical disorders, such as chronic illnesses, physical and functional impairments, frailty, cognitive decline, and dementia [3]. Among the health conditions, cognitive decline is an important mental illness that warrants special care insofar as it might represent a preliminary clinical sign of dementia [4]. Therefore, it is essential to comprehend the etiology of cognitive decline so that a health policy can be developed to prevent cognitive impairment and reduce the likelihood that it will progress to dementia.

Comorbidities, which are defined as the coexistence of one or more chronic illnesses or diseases, are more common in older people and can raise the possibility of functional restrictions and/or limitations in both the physical and mental spheres [5]. Additionally, comorbidities are linked to a higher risk of cognitive decline, which may be connected to the pathophysiology and/or development of cognitive impairments and dementia [6], though the precise pathology underlying the documented association between exposures and outcomes is still poorly understood. From an etiological standpoint, it is well known that lifestyle risk factors, such as obesity, inactivity, frailty, excessive drinking, cigarette use, poor nutritional status, and others, are linked to a higher risk of having comorbidities in older adults [7].

Physical activity (PA) refers to any skeletal-muscle movement that requires a significant energy expenditure above that of rest. As a part of muscular fitness, muscle strength is influenced by the interaction of inherited and environmental factors [8]. It is well established that PA, including aerobic exercise, is crucial for enhancing cognitive function in people of all ages, possibly through increased cerebral blood flow and heightened neuromuscular activity [9,10]. Muscular strength is linked with better cognitive function among older adults. Resistance training has a favorable impact on cognitive ability, possibly via alterations in cerebral blood flow and neuronal activity of the central nervous system [11]. From a therapeutic standpoint, therefore, PA and muscle strength are two distinct factors that positively influence cognitive function [12].

Previous research has found links between cognitive function and PA and physical fitness. For example, the positive association between PA and cognitive function has been reported in previous studies involving Korean and other older Asian adults [13,14,15]. Likewise, lower-extremity muscular strength was an independent predictor of higher cognitive function in older adults who participated in the 1999–2002 National Health and Nutrition Examination Survey [16]. Handgrip strength and 6 min walk were two independent predictors of cognitive function in community-dwelling older Chinese adults [17]. Furthermore, it is probable that those who are physically strong may also be more prone to engaging in regular PA and vice versa, underscoring the importance of considering both as determinants of cognitive function.

Later in life, cognitive function, comorbidities, physical activity, and muscular strength all become intertwined. The nature of the relationships between these four variables, however, is unknown. In this light, the current study sought to investigate the relationships between cognitive function, comorbidities, physical activity, and lower-body muscle strength (LBMS). We hypothesized that PA and LBMS would both act as moderators in determining the impact of comorbidities on cognitive function in elderly Koreans.

## 2. Materials and Methods

### 2.1. Study Design, Setting, and Participants

This is a population-based cross-sectional study. Data for this study were taken from the 2020 Korea Longitudinal Study on Aging (KLoSA), which is a nationwide panel survey of the Korean population conducted by the Korea Employment Information Service (KEIS) in 2020 on behalf of the South Korean Ministry of Employment and Labor. The KEIS carried out the 2020 KLoSA survey across the country using a multi-stage, stratified sampling based on geographical areas and housing types. The 2020 KLoSA survey included a total of 10,097 older individuals aged 65 and older (6062 females and 4035 men) in 15 metropolitan cities and provinces of South Korea. The data collection was conducted using a computer-assisted personal interview. Other detailed information, such as weighting data and imputation methods for non-response, is available elsewhere (https://survey.keis.or.kr/klosa/klosa01.jsp) (accessed on 10 November 2022).

### 2.2. Variables

#### 2.2.1. Cognitive Function

Cognitive function was assessed using the Mini-Mental Status Examination (K-MMSE), which was designed for dementia screening with a maximum total score of 30 points. The presence of cognitive impairment was determined using the previously described age- and education-specific cutoff points that were developed to identify individuals with cognitive impairments for dementia screening [18].

#### 2.2.2. Comorbidities

Comorbidities were assessed based on the presence of at least one of the 25 chronic conditions listed in the other nationwide survey study [19]. The presence of a physician-diagnosed chronic condition(s) was determined by a self-reported questionnaire. Multimorbidity was defined as the presence of two or more chronic diseases.

#### 2.2.3. Physical Activity and Lower-Body Muscle Strength

A self-reported questionnaire that inquired about participation in three domains of PA (i.e., occupational, transportation, and leisure–recreation) lasting at least 10 min per session was used to measure PA. For answers of “yes,” he or she was next prompted to record his or her weekly PA in terms of frequency and duration. The PA volume was then estimated by multiplying the weekly frequency by the duration of each session (minutes). In accordance with the global recommendations, PA was then categorized as being either sufficient (≥150 min per week) or insufficient (no PA or <150 min per week) [20].

A sit-to-stand test (STST), which was detailed elsewhere [21] and reported to have validity and reliability in elderly Koreans [22], was used to evaluate LBMS. Briefly, the participants were told to stand up from a chair 5 times as quickly as they could with both arms crossed over their chest. The performance was graded according to its completion (1 = successfully accomplished, 2 = attempted but failed to complete, and 3 = unable to perform at all). For the purposes of this study, a successfully completed STST was classified as normal, and attempts that could not be performed at all or could not be completed were combined and classified as weak.

#### 2.2.4. Covariates

The covariates were age (years), gender (male vs. female), body mass index (kg/m^2^), marital status, education (elementary, middle/high school, or college), smoking (current/past smoker or non-smoker), and alcohol consumption (0, 1–6, 7 times/week).

### 2.3. Statistics

The normality of the data distribution and multicollinearity were checked using QQ plotting and the variance of the inflation component, respectively. The Student’s t-test and the chi-square test were applied to compare the cognitive function-based subgroups for continuous and categorical variables, respectively. Linear-regression analysis was conducted to estimate standardized coefficients of measured parameters in determining their relationships with cognitive function. Dummy codes were used for categorical variables such as gender (male = 1 and female = 0) and marriage (married with spouse = 1 and all others = 0). The odds ratios (ORs) and 95% confidence intervals (CIs) of cognitive impairment by PA and LBMS were determined using binary logistic regression. Finally, as depicted in Figure 1, Andrew Hayes’ PROCESS macro with Model 2 was used to test the moderating effects of PA (moderator 1, W) and LBMS (moderator 2, Z) on the relationship between comorbidities (independent variable, X) and cognitive performance (dependent variable, Y). The statistical significance of the model was determined with 95% CIs and bias-corrected bootstrapping (*n* = 10,000). All additional statistical significances were evaluated using SPSS-PC version 29.0 at a level of = 0.05. (IBM Corporation, Armonk, NY, USA).

## 3. Results

Table 1 presents the descriptive statistics of the study participants by comorbidity. Those who had multimorbidity were likely to be older (*p* < 0.001), heavier (*p* < 0.001), live alone (*p* < 0.001), be smokers (*p* < 0.001), drink more frequently (*p* < 0.006), be less physically active (*p* < 0.001), have weaker LBMS (*p* < 0.001), have a lower cognitive function (*p* < 0.001), and have a higher rate of cognitive impairment (*p* < 0.001).

Table 2 presents the beta coefficients of linear regression for cognitive function. Cognitive function was inversely associated with age (*p* < 0.001), females (*p* < 0.001), smoking (*p* < 0.001), and multimorbidity (*p* < 0.001) and positively with married with a spouse (*p* < 0.001), BMI (*p* < 0.001), education (*p* < 0.001), alcohol intake (*p* < 0.001), PA (*p* < 0.001), and LBMS (*p* < 0.001). Although the positive correlation coefficient of alcohol consumption is surprising, it may reflect a gender difference in the association between alcohol consumption and cognitive function because the correlation coefficient for men (β = 0.088, *p* = 0.067) is not statistically significant whereas the correlation coefficient (β = 0.403, *p* < 0.001) for women is. In addition, the cognitive benefit of women might be associated with moderate alcohol consumption but not with heavy alcohol consumption [23]. In addition, the overall prevalence of chronic diseases was in the order of hypertension (56.8%), diabetes (24.2%), dyslipidemia (17.1%), rheumatic arthritis (16.5%), osteoarthritis (8.5%), heart diseases (4.5%), angina/myocardial infarction (4.4%), stroke (4.3%), thyroid disease (3.3%), and others.

Table 3 represents the estimated risks for cognitive impairment by PA and LBMS. Multimorbidity was correlated with a higher risk of cognitive impairment (OR = 1.321, 95% CI = 1.167~1.496, *p* < 0.001), which remained statistically significant (OR = 1.415, 95% CI = 1.154~1.736, *p* < 0.001) even after adjustments for age, gender, body mass index, marriage, education, smoking, and alcohol consumption. Insufficient PA was correlated with a higher risk of cognitive impairment (OR = 1.325, 95% CI = 1.219~1.441, *p* < 0.001) compared to sufficient PA, which remained statistically significant (OR = 1.340, 95% CI = 1.160~1.547, *p* < 0.001) even after adjustments for all the covariates. Weak LBMS was correlated with a higher risk of cognitive impairment (OR = 3.240, 95% CI = 2.607~4.027, *p* < 0.001) compared to normal LBMS, which remained statistically significant (OR = 1.719, 95% CI = 1.380~2.143, *p* < 0.001) even after adjustments for all the covariates.

Table 4 displays the relationship between comorbidities (X) and cognitive function (Y) moderated by PA (W) and LBMS (Z). PA moderates the impact of comorbidities on cognitive function (β = −0.3753, 95% CI = −0.5165 to −0.2341), which remained statistically significant (β = −0.3833; 95% CI = −0.4743 to −0.2023) even after adjusting for all the covariates. As shown in Figure 2, individuals with insufficient PA had a substantially steeper slope for the link between comorbidities and cognitive function compared to individuals with sufficient PA. That is, individuals with insufficient PA were likely to experience a more severe influence of chronic diseases on cognitive decline as compared to people with sufficient PA. Likewise, LBMS moderates the impact of comorbidities on cognitive function (β = −0.2353, 95% CI = 0.0774 to 0.3932). When all covariates were controlled for, the moderating effect of LBMS was no longer significant, despite remaining a significant predictor of cognitive function, implying that weak LBMS was an independent predictor of cognitive function regardless of comorbidities.

## 4. Discussion

This study examined the combined effects of PA and LBMS on the relationship between comorbidities and cognitive function in elderly Koreans. In this study population, cognitive function was inversely correlated with comorbidities and positively with PA and LBMS. It was especially interesting to note that having sufficient PA reduced the negative impact of comorbidities on cognitive function. However, muscle strength was found to be an independent predictor of cognitive function rather than a modulator in determining the impact of comorbidities on cognitive function.

In agreement with the current findings, previous studies reported inverse associations between comorbidities and cognitive function in community-dwelling older Korean adults [24], elderly Indian residents [25], and community-dwelling older Irish adults [26]. Furthermore, the negative impact of comorbidities on cognitive function has been observed in Canadian patients with dementia, mild cognitive impairment, and normal cognition [27]; European patients with type-2 diabetes [28]; Chinese patients with first-ever ischemic stroke [29]; Spanish patients with Parkinson’s disease [30]; and older Chinese patients with dementia [31]. Together, the findings from the present study and earlier ones concur that comorbid conditions have an adverse effect on cognitive function among older adults.

Nevertheless, comorbidities and their relationship with cognitive decline might be bi-directional [32]. By analyzing data obtained from community-dwelling adults in 2015–2017 through the Behavioral Risk Factor Surveillance System, the work of Taylor et al. [33], for example, showed that people with subjective cognitive decline were more likely to have a higher prevalence of chronic diseases, including coronary heart disease, stroke, diabetes, asthma, chronic obstructive pulmonary disease, cancer, arthritis, or kidney disease in comparison to people without subjective cognitive decline. In a similar vein, by conducting a secondary prevention study with 200 men aged 57.3±6.3 years, the work of Lutski et al. [34] showed that subjects with two or more chronic conditions had a greater probability of experiencing subjective cognitive impairment over a 20-year follow-up period. Taken together, it appears that comorbidities result in physical and functional limitations or changes in brain macrostructure and microstructure, which have a negative impact on cognitive function [35,36] or vice versa [37,38]. As a result, a prospective cohort study will be required to determine the precise direction of the relationship between comorbidities and cognitive function.

The cognitive benefit of PA observed in the current study has been reported in previous studies involving older Korean adults. For instance, Song and Park [39] examined changes in PA and its relationship with cognitive function using baseline and follow-up data from the KLoSA and showed that older adults who became or remained inactive had a higher risk of cognitive impairment compared to their counterparts who remained physically active. By analyzing the data obtained from 864 community-dwelling older adults from the Suwon Geriatric Mental Health Center, the work of Kim et al. [13] showed that PA positively contributed to cognitive function, perhaps via anti-depressant effects, in study subjects. Additionally, findings from the Longitudinal Ageing Study in India [14], French tri-city research [15], and the Mayo Clinic Study of Aging [40] also recognize the cognitive benefits of PA among community-dwelling older adults. Taken together, the findings from the current and previous studies support the cognitive benefits of PA later in life.

Additionally, muscular strength is just as important as PA in reducing the physical- and mental-health burden of chronic disorders, such as depression [41] and cognitive decline [36]. For example, by examining data taken from the KLoSA 2006–2015 and 2014–2018, respectively, Jeong and Kim [42] and Lee et al. [43] demonstrated that low handgrip strength was positively related to an elevated risk of cognitive impairment among older Korean adults. Similarly, Frith and Loprinzi [16], by evaluating data taken from the 1999–2002 National Health and Nutrition Examination Survey, examined the relationship between lower-extremity muscle strength and cognitive performance in a representative sample of older persons. That study shows that people with high muscle strength had better cognitive function than people with low muscle strength. Taken together, the findings from the current and previous studies strengthen our understanding of the cognitive benefits of muscular strength among older adults, regardless of comorbidities.

Although handgrip strength is more commonly tested in older adults, no sufficient evidence is available to support a definitive tool for measuring overall muscular strength and physical function [44]. Furthermore, other research suggests that lower-limb muscular strength is a better predictor of overall muscle strength or an alternative for people with hand disabilities [45]. The STST used in this study is yet another proxy for lower-extremity strength and muscular performance. A future study should look into the advantages and disadvantages of these surrogate measures of muscular strength. On the other hand, we believe that this is the first study to describe the moderating effect of sufficient PA in determining the association between comorbidities and cognitive function in elderly Asians, including Koreans. Therefore, the current findings of the study extend the cognitive benefits of PA reported in previous studies by demonstrating that regular PA may help older adults prevent and/or mitigate cognitive decline caused by chronic diseases.

The cognitive benefit of sufficient PA observed in the current study can be explained by several factors. First, having one or more chronic conditions may lead to an increased risk of depression, which may negatively contribute to cognitive function in older adults [46]. In contrast, sufficient PA has antidepressant effects that may attenuate the negative impact of comorbidities on cognitive function in older adults [47]. Second, the cognitive benefits of PA may be mediated via anti-inflammatory effects that may function to attenuate the impact of comorbidities on cognitive function [48]. Third, PA may function together to counteract cognitive decline associated with comorbidities via neuroplasticity, angiogenesis, neurogenesis, and mitochondrial biogenesis [49]. Lastly, PA is associated with better physical function [50], better health-associated quality of life [51], and emotional and physical well-being [52], which positively contribute to cognitive function among older adults.

The study has limitations. We are unable to propose a cause-and-effect explanation for the current findings due to the cross-sectional character of the study. Second, there may be a reciprocal association between comorbidities, PA, muscular strength, and cognitive performance, which remains to be confirmed in a subsequent study. Third, although the MMSE was used to assess cognitive function in this study, having additional reliable measures such as the Montreal Cognitive Assessment would be preferable in assessing various cognitive domains. Fourth, the assessment of PA using a self-reported questionnaire may either underestimate sedentary behaviors or overestimate moderate and high-intensity PAs among older adults [53]. Therefore, an objective measurement of PA using an accelerometer would be necessary for a future study.

## 5. Conclusions

This study examined the relationships between cognitive function, comorbidities, PA, and LBMS in older Korean adults. We found that PA is a modulator in determining the impact of comorbidities on cognitive function, and LBMS is an independent determinant of cognitive function, regardless of comorbidities. Therefore, the current findings of the study imply the clinical importance of having both sufficient PA and normal LBMS for a therapeutic strategy against cognitive decline associated with chronic diseases.

## Figures and Tables

**Figure 1 behavsci-13-00212-f001:**
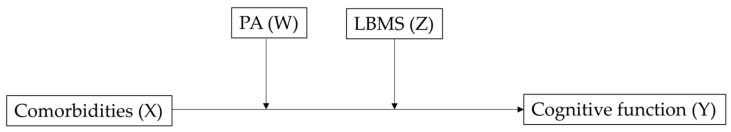
An illustration of the relationship between comorbidities (X) and cognitive function (Y) moderated by physical activity (W) and lower-body muscle strength (Z).

**Figure 2 behavsci-13-00212-f002:**
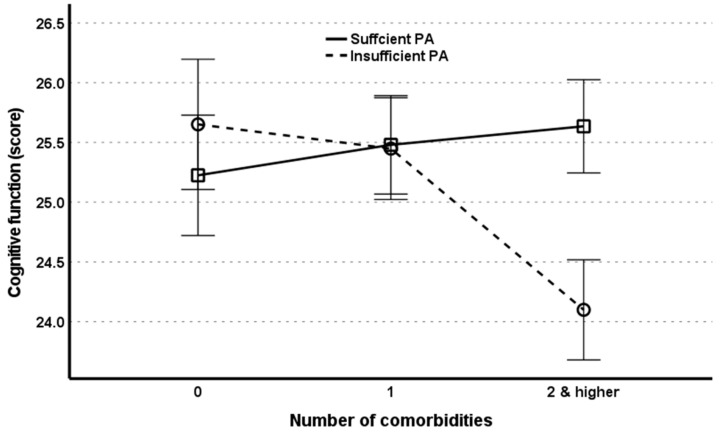
Effect of physical activity (PA) on the relationship between comorbidities and cognitive function.

**Table 1 behavsci-13-00212-t001:** Descriptive statistics of study participants by comorbidity.

Variables	Number of Comorbidities			*p* for Linear Trends
	0 (*n* = 1686/16.7%)	1 (*n* = 2956/29.3%)	2 (*n* = 5445/54.0%)	
Age in years, mean (95% CI)	70.9 (70.6~71.2)	72.6 (72.4~72.8)	74.9 (74.7~75.1)	<0.001
BMI in kg/m^2^, mean (95% CI)	23.3 (23.2~23.4)	23.5 (23.4~23.6)	23.7 (23.6~23.8)	<0.001
Gender, *n* (%)				<0.001
Male	831 (49.3)	1326 (44.9)	1878 (34.4)	
Female	855 (50.7)	1630 (55.1)	3577 (65.6)	
Marriage status				<0.001
Never married	13 (0.8)	5 (0.2)	25 (0.5)	
Married with a spouse	1179 (69.9)	1891 (64.0)	2861 (52.4)	
Married without a spouse	494 (29.3)	1060 (35.9)	2569 (47.1)	
Educational background, *n* (%)				<0.001
Elementary or less	527 (31.3)	1133 (38.3)	2888 (52.9)	
Middle/high school	1044 (61.9)	1644 (55.6)	2349 (43.1)	
College or higher	115 (6.8)	179 (6.1)	218 (4.0)	
Smoking status, *n* (%)				<0.001
Current/past smokers	234 (13.9)	363 (12.3)	507 (9.3)	
Alcohol intake (times/week)				0.006
0	-	-	-	
1–6	619 (81.1)	1058 (89.0)	1483 (87.2)	
≥7	144 (18.9)	131 (11.0)	218 (12.8)	
Physical-activity status, *n* (%)				<0.001
Sufficient	957 (56.8)	1540 (52.1)	2745 (50.3)	
Insufficient	729 (43.2)	1416 (47.9)	2710 (49.7)	
Lower-body muscle strength, *n* (%)				<0.001
Normal	1469 (91.5)	2336 (83.0)	3519 (67.8)	
Weak	136 (8.5)	479 (17.0)	1672 (32.2)	
Cognitive function, mean (95% CI)	25.3 (25.1~25.6)	24.9 (24.8~25.1)	23.7 (23.5~23.8)	<0.001
^#^ Cognitive impairment, *n* (%)	493 (29.4)	915 (31.4)	1905 (36.0)	<0.001

Physical activity status; sufficient was defined as meeting the WHO recommendations for PA (or at least 150 min per week). Weak lower-body muscle strength was defined as no attempt or failure to complete five times of sit-to-stand tests. ^#^ Age-, gender-, and education-specific rate of cognitive impairment.

**Table 2 behavsci-13-00212-t002:** Beta coefficients of simple linear regression for cognitive function.

Variables	Unstandardized Beta	Standardized Beta	95% CI	*p*-Value
Age	−0.281	−0.346	−0.296~−0.266	<0.001
Gender	1.306	0.121	1.094~1.519	<0.001
Marriage	1.185	0.168	1.605~2.205	<0.001
Body mass index	0.100	0.049	0.060~0.140	<0.001
Education	0.447	0.337	0.423~0.472	<0.001
Smoking	−1.496	−0.088	−1.831~−1.162	<0.001
Alcohol intake	0.221	0.091	0.143~0.299	<0.001
Comorbidities	−0.535	−0.148	−0.996~−0.705	<0.001
PA	−1.482	−0.139	−1.690~−1.275	<0.001
LBMS	−4.215	−0.338	−4.452~−3.977	<0.001

CI: confidence interval; PA: physical activity; LBMS: lower-body muscle strength.

**Table 3 behavsci-13-00212-t003:** Odds ratios (ORs) and 95% confidence intervals (CIs) of cognitive impairment by levels of physical activity (PA) and lower-body muscle strength (LBMS).

Predictors	Model 1		Model 2	
	OR (95% CI)	*p*-Value	OR (95% CI)	*p*-Value
PA				
Sufficient	1 (reference)		1 (reference)	
Insufficient	1.325 (1.219~1.441)	<0.001	1.340 (1.160~1.547)	<0.001
LBMS				
Normal	1 (reference)		1 (reference)	
Weak	3.240 (2.607~4.027)	<0.001	1.719 (1.380~2.143)	<0.001
Number of comorbidities				
0	1 (reference)		1 (reference)	
1	1.099 (0.964~1.253)	0.159	1.063 (0.864~1.307)	0.564
≥2	1.321 (1.167~1.496)	<0.001	1.415 (1.154~1.736)	<0.001

Model 1: unadjusted. Model 2: adjusted for age, gender, body mass index, marriage, education, smoking, and alcohol consumption.

**Table 4 behavsci-13-00212-t004:** A moderation analysis of physical activity and lower-body muscle strength on the relationship between comorbidity and cognitive function.

Predictors	Coefficients	SE	t	*p*	95% CI	
					Lower	Upper
Model 1 (R^2^ = 0.1277, F = 275.7519, *p* < 0.001)						
Comorbidities (X)	0.2443	0.1079	2.2631	0.0237	0.0327	0.4559
Physical activity (W)	−0.2432	0.1655	−1.4700	0.1416	−0.5675	0.0811
Interaction 1 (X × W)	−0.3753	0.0720	−5.2092	<0.001	−0.5165	−0.2341
LBMS (Z)	−4.2866	0.2163	−19.8182	<0.001	−4.7106	−3.8626
Interaction 2 (X × Z)	0.2353	0.0805	2.9210	0.035	0.0774	0.3932
R^2^ change due to X × W = 0.0025 (F = 27.1362, *p* < 0.001) R^2^ change due to X × Z = 0.0008 (F = 6.5323, *p* = 0.0035) R^2^ change due to both = 0.0028 (F = 15.0746, *p* < 0.001)						
Model 2 (R^2^ = 0.1941, F = 226.4264, *p* < 0.001)						
Comorbidities (X)	0.3852	0.1042	3.6962	0.002	0.1809	0.5895
Physical activity (W)	−0.0019	0.1596	−0.0119	0.991	−0.3148	0.3110
Interaction 1 (X × W)	−0.3383	0.0694	−4.8758	<0.001	−0.4743	−0.2023
LBMS (Z)	−2.5078	0.2199	−11.4042	<0.001	−2.9388	−2.0767
Interaction 2 (X × Z)	0.0647	0.0778	0.8309	0.406	−0.0879	0.2172
R^2^ change due to X × W = 0.0020 (F = 23.7736, *p* < 0.001) R^2^ change due to X × Z = 0.0001 (F = 0.6904, *p* = 0.406) R^2^ change due to both = 0.0020 (F = 11.9443, *p* < 0.001)						

Model 1 unadjusted. Model 2 adjusted for age, sex, body mass index, marriage, education, smoking, and alcohol consumption. SE: standard error; CI: confidence interval. LBMS: lower-body muscle strength; X = moderator 1; Z = moderator 2.

## Data Availability

The datasets used and analyzed during this study are available from the national public database (survey.keis.or.kr/eng/myinfo/login.jsp) (accessed on 10 November 2022).

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
