# Peer review of "Cognitive Function, and Its Relationships with Comorbidities, Physical Activity, and Muscular Strength in Korean Older Adults"

_behavsci, 2023, doi:10.3390/bs13030212_

Round 1
Reviewer 1 Report
The main objective of this study is about to examine the combined effects of PA and LBMS on the relationship between comorbidities and cognitive function in elderly Koreans. The topic is interesting, however, need some improvement.
1. In introduction, I can see just previous studies and the objective of the study. I haven’t seen any gaps from previous studies.
2. From the beginning I can answer the research questions, so please bring more evidence why we need such research?
3. What is the novelty of this study?
4. How authors access to the database?
5. It is better the authors come up with research hypotheses section and prepare every hypothesis based on their background of the study.
6. Tables 1,2, and 3 are not in order in the text.
7. In Table 2, interpretation 95% CI of gender and marriage is not clear for me. These two variables are nominal variables. How the authors interpret for example 95% CI for gender ( -1.519 ~ -1.094)?
8. Based on Table 2, alcohol consumption has a positive and significant impact. It would be interesting output if the authors interpret more about this part of their study.
9. Is it possible in Table 2 to put standardized beta also?
10. Section 5. Conclusion is not completed yet. Please rewrite this section properly.
Author Response
In our Responses to the Comments and Suggestions by Reviewer #1
We deeply appreciate the reviewers for their thoughtful comments. We did our best to address all the comments/critics point-by-point, which are highlighted in yellow color. Two references are added, and one reference is removed, and they are listed on the last page of our responses.
Questions 1,2,3,5)
Q1. In introduction, I can see just previous studies and the objective of the study. I haven’t seen any gaps from previous studies.
Q2. From the beginning I can answer the research questions, so please bring more evidence why we need such research?
Q3. What is the novelty of this study?
Q5. It is better the authors come up with research hypotheses section and prepare every hypothesis based on their background of the study.
ANS1,2,3,5) In our responses to questions of 1, 2, 3, and 5, we revised and rewritten the Introduction to better describe the study purpose, research questions, novelty, and research hypothesis.
“Physical activity (PA) refers to any skeletal muscle movement that requires a significant energy expenditure above that of rest. As a part of muscular fitness, muscle strength is influenced by the interaction of inherited and environmental factors [8]. It is well established that PA, including aerobic exercise, is crucial for enhancing cognitive function in people of all ages, possibly through increased cerebral blood flow and heightened neuromuscular activity [9, 10]. Muscular strength is linked with better cognitive function among older adults. Resistance training has a favorable impact on cognitive ability, possibly via altering alterations in cerebral blood flow and neuronal activity of the central nervous system [11]. From a therapeutic standpoint, therefore, PA and muscle strength are two distinct factors that positively influence cognitive function [12].
Previous research has found links between cognitive function and PA and physical fitness. For example, the positive association between PA and cognitive function has been reported in previous studies involving Korean and other Asian older adults [13-15]. Likewise, lower-extremity muscular strength was an independent predictor of higher cognitive function in older adults who participated in the 1999-2002 National Health and Nutrition Examination Survey [16]. Handgrip strength and 6-min walk were two independent predictors of cognitive function in community-dwelling Chinese older adults [17]. Furthermore, it is probable that those who are physically strong may also be more prone to engage in regular PA, and vice versa, underscoring the importance of considering both as determinants of cognitive function.
Later in life, cognitive function, comorbidities, physical activity, and muscular strength all become intertwined. The nature of the relationships between these four variables, however, is unknown. In this light, the current study sought to investigate the relationships between cognitive function, comorbidities, physical activity, and lower-body muscle strength (LBMS). We hypothesized that PA and LBMS would both act as moderators in determining the impact of comorbidities on cognitive function in elderly Koreans.”
Q4) How authors access to the database?
ANS4) Thanks for the comment. The KLoSA data can be downloaded from the Korea Employment Information Service (https://survey.keis.or.kr/klosa/klosa01.jsp). We accessed the database via the website on November 10, 2022.
Q6) Tables 1,2, and 3 are not in order in the text.
ANS6) Thanks for the comment. The order of Tables 1-3 has been corrected.
Q7) In Table 2, interpretation 95% CI of gender and marriage is not clear for me. These two variables are nominal variables. How the authors interpret for example 95% CI for gender ( -1.519 ~ -1.094)?
ANS7) Thanks for the thoughtful comment. In response to the comment, we decided to remove the nominal data from the regression analysis.
Q8) Based on Table 2, alcohol consumption has a positive and significant impact. It would be interesting output if the authors interpret more about this part of their study.
ANS8) Thanks for the comment. It is unexpected and surprising. In general, Korean older men have a higher rate of alcohol consumption than Korean older women. Older men have higher MMSE scores than older women. Therefore, the positive beta of alcohol consumption may reflect a gender difference in cognitive function. When we re-run the regression analysis separately by gender, the correlation coefficient for men was not statistically significant (beta=0.088 and p=0.067), while the correlation coefficient for women was statistically significant (beta=0.403 and p<0.001).
In our response to the comment, we added the following statement in the Result.
“Although the positive correlation coefficient of alcohol consumption is surprising, it may reflect a gender difference in the association between alcohol consumption and cognitive function because the correlation coefficient for men (β =0.088, p=0.067) is not statistically significant while the correlation coefficient (β=0.403, p<0.001) for women is. In addition, the cognitive benefit of women might be associated with moderate alcohol consumption but not with heavy alcohol consumption [24].”
Q9) Is it possible in Table 2 to put standardized beta also?
ANS9) Thanks. Standardized beta values are added to Table 2 (refer to Table 2).
Q10) Section 5. Conclusion is not completed yet. Please rewrite this section properly.
ANS10) Thanks for the comment. We are sorry for this. We added “Conclusion” as follows:
“This study examined the relationships between cognitive function, comorbidities, PA, and LBMS in Korean older adults. We found that PA is a modulator in determining the impact of comorbidities on cognitive function, and LBMS is an independent determinant of cognitive function, regardless of comorbidities. Therefore, the current findings of the study imply the clinical importance of having both sufficient PA and normal LBMS for a therapeutic strategy against cognitive decline associated with chronic diseases.”
List of Added References
- Lyu, J.; Lee, S.H. Alcohol consumption and cognitive impairment among Korean older adults: does gender matter? Int Psychogeriatr. 2014, 26, 335-340.
- Zhao, X.; Huang, H.; Du, C. Association of physical fitness with cognitive function in the community-dwelling older adults. BMC Geriatr. 2022, 22, 868.
List of Deleted References
- Bennie, J.A.; De Cocker, K.; Teychenne, M.J.; Brown, W.J.; Biddle, S.J.H.The epidemiology of aerobic physical activity and muscle-strengthening activity guideline adherence among 383,928 U.S. adults. Int J Behav Nutr Phys Act. 2019, 16, 34.

Reviewer 2 Report
The author states that the purpose of the study was to investigate and evaluate the viability of a physical activity program.
The introduction/literature review sections need to be restructured
The information in the introduction section is too short and the result are in terms of scientific evidence
Some information in the result sections corresponds to the introduction section
I will suggest that the author consider including analyzes of psychological variables
Author Response
In our Responses to the Comments and Suggestions by Reviewer #2
We deeply appreciate the reviewers for their thoughtful comments. We did our best to address all the comments/critics point-by-point, which are highlighted in yellow color. Two references are added, and one reference is removed, and they are listed on the last page of our responses.
Q1) The introduction/literature review sections need to be restructured. The information in the introduction section is too short and the result are in terms of scientific evidence. Some information in the result sections corresponds to the introduction section
ANS1) Thanks for the comments and suggestions. In our responses to the comments, we rewritten the Introduction as follows:
“Physical activity (PA) refers to any skeletal muscle movement that requires a significant energy expenditure above that of rest. As a part of muscular fitness, muscle strength is influenced by the interaction of inherited and environmental factors [8]. It is well established that PA, including aerobic exercise, is crucial for enhancing cognitive function in people of all ages, possibly through increased cerebral blood flow and heightened neuromuscular activity [9, 10]. Muscular strength is linked with better cognitive function among older adults. Resistance training has a favorable impact on cognitive ability, possibly via altering alterations in cerebral blood flow and neuronal activity of the central nervous system [11]. From a therapeutic standpoint, therefore, PA and muscle strength are two distinct factors that positively influence cognitive function [12].
Previous research has found links between cognitive function and PA and physical fitness. For example, the positive association between PA and cognitive function has been reported in previous studies involving Korean and other Asian older adults [13-15]. Likewise, lower-extremity muscular strength was an independent predictor of higher cognitive function in older adults who participated in the 1999-2002 National Health and Nutrition Examination Survey [16]. Handgrip strength and 6-min walk were two independent predictors of cognitive function in community-dwelling Chinese older adults [17]. Furthermore, it is probable that those who are physically strong may also be more prone to engage in regular PA, and vice versa, underscoring the importance of considering both as determinants of cognitive function.
Later in life, cognitive function, comorbidities, physical activity, and muscular strength all become intertwined. The nature of the relationships between these four variables, however, is unknown. In this light, the current study sought to investigate the relationships between cognitive function, comorbidities, physical activity, and lower-body muscle strength (LBMS). We hypothesized that PA and LBMS would both act as moderators in determining the impact of comorbidities on cognitive function in elderly Koreans.”
Q2) I will suggest that the author consider including analyzes of psychological variables.
ANS2) Thanks for the comments. The purpose of the current study was to examine the nature of the relationships between cognitive function, comorbidities, physical activity, and lower-body muscle strength. But we hope to revisit the issue involving psychological variables too.
List of Added References
- Lyu, J.; Lee, S.H. Alcohol consumption and cognitive impairment among Korean older adults: does gender matter? Int Psychogeriatr. 2014, 26, 335-340.
- Zhao, X.; Huang, H.; Du, C. Association of physical fitness with cognitive function in the community-dwelling older adults. BMC Geriatr. 2022, 22, 868.
List of Deleted References
- Bennie, J.A.; De Cocker, K.; Teychenne, M.J.; Brown, W.J.; Biddle, S.J.H.The epidemiology of aerobic physical activity and muscle-strengthening activity guideline adherence among 383,928 U.S. adults. Int J Behav Nutr Phys Act. 2019, 16, 34.

Reviewer 3 Report
Cognitive Function and its Relationships with Comorbidities, Physical Activity, and Muscular Strength in Older Adults is a vital topic to be investigated due to its importance for healthy and active aging. Therefore, the study needs to present more clearly methodological aspects and include its modification in the Other manuscript sections. Elements that need to be considered:
1 – Structure the manuscript based on strobe recommendations (doi: 10.4103/sja.SJA_543_18), strengthening the methods describing more detail such as study design type, Analytical topics used to assess the complexity of moderation effect (independent Variable, levels of the moderator, for example).
2- It is not clear the QUality control used to assess the variables of interest;
3 –Subject selection criteria and variable stratification (levels to Analyse lower-body muscle strength - LBMS) Why comment that muscular strength is a better predictor of overall muscle strength once the authors decided to use another indicator? Present this topic in the discussion.
Author Response
In our Responses to the Comments and Suggestions by Reviewer #3
We deeply appreciate the reviewers for their thoughtful comments. We did our best to address all the comments/critics point-by-point, which are highlighted in yellow color. Two references are added, and one reference is removed, and they are listed on the last page of our responses.
Q1) Structure the manuscript based on strobe recommendations (doi: 10.4103/sja.SJA_543_18), strengthening the methods describing more detail such as study design type, Analytical topics used to assess the complexity of moderation effect (independent Variable, levels of the moderator, for example).
ANS1) Thanks for the comment. In the Methods, “Study design” is added. “Measurements” are renamed as “Variables”. “X (comorbidities)” is renamed as “Independent Variable” and Y (cognitive function) is renamed as “Dependent in Statistics”.
Q2) It is not clear the QUality control used to assess the variables of interest;
ASN2) Thanks for the comment. As mentioned in Materials and Methods (Data source), KLoSA is a nationwide survey conducted by the Korea Employment Information Service (KEIS). A complete description of the survey design and data collection including quality control can be obtained at https://survey.keis.or.kr/eng/klosa/klosa01.jsp.
Q3) Subject selection criteria and variable stratification (levels to Analyse lower-body muscle strength - LBMS) Why comment that muscular strength is a better predictor of overall muscle strength once the authors decided to use another indicator? Present this topic in the discussion.
ANS3) Thanks for the comment. To simplify the issue, we removed the following sentence.
“Older adults are more frequently tested for handgrip strength [21], but lower limb muscular strength is a better predictor of overall muscle strength.”
List of Added References
- Lyu, J.; Lee, S.H. Alcohol consumption and cognitive impairment among Korean older adults: does gender matter? Int Psychogeriatr. 2014, 26, 335-340.
- Zhao, X.; Huang, H.; Du, C. Association of physical fitness with cognitive function in the community-dwelling older adults. BMC Geriatr. 2022, 22, 868.
List of Deleted References
- Bennie, J.A.; De Cocker, K.; Teychenne, M.J.; Brown, W.J.; Biddle, S.J.H.The epidemiology of aerobic physical activity and muscle-strengthening activity guideline adherence among 383,928 U.S. adults. Int J Behav Nutr Phys Act. 2019, 16, 34.

Round 2
Reviewer 1 Report
The authors amended all of my comments in their manuscript. Thank you and wish you all the best!
Author Response
Thanks much!
Reviewer 3 Report
The authors incorporate the recommendations which improved the analysis and results. Considering that, I recommend that They include in the abstract the details pointed out in the methods section.
Author Response
In our Responses to the Comments and Suggestions by Reviewer #2
We deeply appreciate the reviewers for their thoughtful comments. We did our best to address all the comments/critics point-by-point, which are highlighted in yellow color.
Q1) The authors incorporate the recommendations which improved the analysis and results. Considering that, I recommend that They include in the abstract the details pointed out in the methods section.
ANS) Thanks for the comment. In our response to the comments, the details of analyses are added in the abstract and the Methods:
Abstract
This is a population-based cross-sectional study. Data for this study were taken from the 2020 Korea Longitudinal Study on Aging (KLoSA) in South Korea using a computer-assisted personal interview. The 2020 KLoSA survey included a total of 10,097 older individuals aged 65 and older (6,062 females and 4,035 men).
2.1. Study design, setting, and participants
This is a population-based cross-sectional study. Data for this study were taken from the 2020 Korea Longitudinal Study on Aging (KLoSA), which is a nationwide panel survey of the Korean population conducted by the Korea Employment Information Service (KEIS) in 2020 on behalf of the South Korean Ministry of Employment and Labor. The KEIS carried out the 2020 KLoSA survey across the country using a multi-stage, stratified sampling based on geographical areas and housing types. The 2020 KLoSA survey included a total of 10,097 older individuals aged 65 and older (6,062 females and 4,035 men) in 15 metropolitan cities and provinces of South Korea. The data collection was conducted using a computer-assisted personal interview. Total quality management was used to manage the data quality (TQM). Other detailed information, such as weighting data and imputation methods for non-response, is available elsewhere (https://survey.keis.or.kr/klosa/klosa01.jsp) (accessed on November 10, 2022).